# Micromagnetic and Robust Evaluation of Surface Hardness in Cr12MoV Steel Considering Repeatability of the Instrument

**DOI:** 10.3390/s23031273

**Published:** 2023-01-22

**Authors:** Zhixiang Xing, Xianxian Wang, Mengshuai Ning, Cunfu He, Xiucheng Liu

**Affiliations:** Faculty of Materials and Manufacturing, Beijing University of Technology, Beijing 100124, China

**Keywords:** micromagnetic testing, repeatability, quantitative prediction, surface harness, robustness

## Abstract

The combination of multifunctional micromagnetic testing and neural network-based prediction models is a promising way of nondestructive and quantitative measurement of steel surface hardness. Current studies mainly focused on improving the prediction accuracy of intelligent models, but the unavoidable and random uncertainties related to instruments were seldom explored. The robustness of the prediction model considering the repeatability of instruments was seldom discussed. In this work, a self-developed multifunctional micromagnetic instrument was employed to perform the repeatability test with Cr12MoV steel. The repeatability of the instrument in measuring multiple magnetic features under both static and dynamic conditions was evaluated. The magnetic features for establishing the prediction model were selected based on the consideration of both the repeatability of the instrument and the ability of magnetic features in surface hardness evaluation. To improve the robustness of the model in surface hardness prediction, a modelling strategy considering the repeatability of the instrument was proposed. Through removing partial magnetic features with higher mean impact values from input nodes, robust evaluation of surface hardness in Cr12MoV steel was realized with the multifunctional micromagnetic instrument.

## 1. Introduction

Quality control is important in the manufacturing process of advanced high-strength steels and high-end ferromagnetic components. Mechanical properties (such as surface hardness, yield strength, and elongation, etc.) are usually measured to evaluate the quality of steel sheets and ferromagnetic components. Traditional measurement ways of mechanical properties, such as indentation-based and tensile test methods, are destructive and not applicable to online tests. In ferromagnetic materials, the microstructures (grain boundary, precipitates, dislocation, etc.) not only hinder the motion of magnetic domains under the action of an external magnetic field but also determine the mechanical behavior under an external load [1]. Intrinsic connections between the magnetic properties and mechanical properties of ferromagnetic materials had been confirmed by numerous experiments [2,3,4,5]. Such phenomenological conclusions suggest the feasibility of magnetic evaluation of mechanical properties in nondestructive ways.

Among all the signatures representing the magnetic properties of ferromagnetic materials, micro-magnetic signals are the indicators of material magnetization on a microscopic scale and mainly originate from the pinning effect of microstructures on magnetic domains. Therefore, micro-magnetic signals (such as magnetic Barkhausen noise and incremental permeability) are sensitive to the variation in micro-structures and consequently mechanical properties [6]. Some features extracted from micro-magnetic signals demonstrated high correlations with mechanical properties. However, the correlation models varied with material type, heat treatment process, and even the used instrument, thus limiting the universality of micro-magnetic evaluation methods [7,8]. Current researches tend to discuss each case separately while modeling the relationship between the magnetic features and mechanical properties is required. For instance, Sorsa et al. [9] collected magnetic Barkhausen noise from samples of case-hardened steel 18CrNiMo7-6 and proposed the data-based approach of establishing multivariable linear regression models to predict residual stress and surface hardness. Dong et al. [10] established three-layer back-propagation neural network (BP-NN) models for the quantitative evaluation of residual stress and surface hardness in deep-drawn parts based on magnetic Barkhausen noise technology. 

Recent studies proved that enriching the parametric space of magnetic features could improve the prediction accuracy of models [11,12]. Therefore, in the proposed multifunctional micromagnetic evaluation method, micromagnetic signals together with other magnetic signatures (such as tangential magnetic field, eddy current, and magnetic hysteresis curve, etc.) were simultaneously measured as input nodes of the model. Jedamski et al. [13] employed the 3MA-II system developed by Fraunhofe-IZFP to realize the quantitative evaluation of hardness and case hardening depth in steel bars of 18CrNiMo7-6. The data from four types of micromagnetic signals were used to train the prediction model. Though the performance of the established model based on an artificial neural network was affected by the teaching steps, the advantages of the neural network method over the methods of linear regression analysis were obvious. Akhlaghi et al. [14] combined the magnetic features extracted from both the magnetic hysteresis loop and eddy current as the input nodes of the generalized regression neural network (GRNN), established the GRNN model to accurately estimate the hardness profile of steel specimens subjected to Jominy test, and found that the calibration of the measurement system (yields as a GRNN model) might be invalid once the hardware or dimensional parameters of the setup changed.

The performance of the multifunctional micromagnetic method combined with an artificial neural network highly relies on the quality of magnetic features measured by instruments and the performances (including accuracy, robustness, etc.) of prediction models. The repeatability of instruments is related to the quality of magnetic features, which may be affected by the random error involved in a single measurement. The sources of random errors include the stochastic process of domain motion, slight fluctuation in electrical parameters of instruments, the minor difference in the contact state between sensors and specimen surface, etc. Improving the prediction accuracy of intelligent models through structural optimization of neural networks has been extensively explored [15,16], but the robustness of the prediction model considering the repeatability of the multifunctional micromagnetic instrument was seldom investigated. If the robustness of the established prediction models is poor, large errors may occur in the quantitative prediction of mechanical properties using micromagnetic testing instruments. 

In this study, attempts were made to find a way for robust prediction model establishment considering the repeatability of the instrument in micromagnetic features measurements. A self-developed multifunctional micromagnetic instrument was employed to perform repeatability tests for the quantitative prediction of surface harness in Cr12MoV steel. With the repeatability testing data collected under both static and dynamic conditions, the repeatability of the instrument in measuring multiple magnetic features was evaluated with the coefficient of variation. Based on the comprehensive consideration of the repeatability of the instrument and the ability of magnetic features in surface hardness evaluation, the magnetic features were filtered for establishing the prediction model based on a feed-forward neural network (FNN). To improve the robustness of FNN models while the instruments suffer random uncertainties, a selection strategy of input nodes is proposed based on the consideration of the balance between influence weight and robustness. The prediction results showed that the robustness of the prediction model could be improved by properly and partially eliminating magnetic features with higher mean impact value (MIV) from input nodes.

The rest of this paper is organized as follows. Detailed information about the experimental set-up and typical experimental results of measured magnetic features are given in Section 2. The repeatability of the instrument under static and dynamic conditions is discussed in Section 3. In Section 4 and Section 5, the prediction model establishment and its robustness evaluation are investigated. The research findings obtained in this work are summarized in the Conclusions.

## 2. Experiments

### 2.1. Experimental Set-Up

Cold work die steel plates of Cr12MoV in Chinese standards were selected for specimen preparation. A batch of rectangular specimens with identical dimensions of 250 × 60 × 3 mm were cut from the as-received steel plates. The specimens were quenched by heating to 1030 °C step-by-step and cooling in nitrogen for 25 min. The quenched specimens were tempered for 210 min at different temperatures. A total of eight specimens were prepared by changing the tempering temperature from 575 °C to 720 °C so that the microstructures and surface hardness of the specimens could be adjusted. The specimen surface was slightly ground to remove the oxide layer for Vickers hardness tests. During hardness tests, the load applied by the indenter remained to be 30 kg. Three randomly selected locations at the central area of the surface were tested. The measured values of Vickers hardness (HV30) are listed in Table 1.

Multifunctional micro-magnetic tests were performed on all eight specimens with a self-developed instrument of MaginFrame. The configuration of the instrument was shown in Figure 1a. The details of the sensor and the operation procedure for the MaginFrame instrument could be found in the previous study [17]. A computer-controlled signal generator was employed to generate two channels of sinusoidal waves. One of them had a frequency range of 50 to 500 Hz and its amplitude was amplified by a voltage-current conversion power amplifier. The amplitude of the amplified current could be adjusted in the range of 1 to 4 A. Another sinusoidal wave with the frequency range of 10 kHz to 1 MHz passed through a current amplifier before being fed into the transmitter coil. The maximum amplitude of the high-frequency sinusoidal current was limited to 100 mA. The magnetization coil wound onto the yoke and the transmitter coil of the air core were deployed to provide low-frequency and high-frequency magnetic fields, respectively. Through controlling the parameters (duration, amplitude, and time delay) of the two-channel sinusoidal waves, intermittently superimposed high- and low-frequency magnetic fields were induced for material magnetization. With this novel magnetization technology, several types of magnetic signals could be generated simultaneously. The compound magnetic signals were measured with combined magnetic sensors (Hall sensor and receiver coil) whose sensitive axes were orthogonal to each other. A total of four types of magnetic signals including the tangential magnetic field (TMF), magnetic Barkhausen noise (MBN), engineer incremental permeability (EIP), and multi-frequency eddy current (MFEC) were measured in a period of single excitation.

Figure 1b demonstrates the whole experiment set-up for multifunctional micro-magnetic tests. The instrument of MaginFrame was integrated with a six-DOF robot system. In the experiment, all the specimens were clamped on a workbench. The moving trajectory of the robotic arm was programmed in a teaching mode so that the sensor held by the robotic arm could test all the specimens in sequence. The six-DOF robot system could suppress the random errors caused by manual operations. During the testing process, the parameters of the current for generating the low-frequency magnetic field were selected as 200 Hz and 2 A in peak amplitude. The amplitude of the high-frequency excitation current was selected as 1 V for EIP and MFEC tests. Four frequencies of 10 kHz, 20 kHz, 50 kHz, and 100 kHz were selected for eddy current analysis. In the EIP test, the high-frequency excitation current had a frequency of 100 kHz. The sampling rate for all the signal acquisition channels was fixed as 2 MSa/s. A total of 41 features (Appendix ATable A1) were extracted from the measured four types of magnetic signals.

### 2.2. Experimental Results

The performances of the developed experimental set-up in measuring the magnetic features and quantitatively predicting the surface hardness of Cr12MoV steel were explored below. The experiments were performed in two stages to collect the required data for evaluating the repeatability of instruments and seeking prediction models of high robustness.

In the first stage of experiments, all eight specimens of different surface hardness were tested under static conditions. After the sensor was in vertical contact with the test location of the specimen, the multifunctional micro-magnetic test was repeated ten times while keeping the sensor steady. Under static conditions, the random errors related to the instrument mainly originate from the stochastic process of domain motion and slight fluctuations in the electrical parameters of the instrument.

In each test, magnetic signals in five magnetization cycles of the low-frequency magnetic field were acquired and averaged. Figure 2 shows the patterns of the multifunctional magnetic signals obtained from the specimens of different surface hardness. The features of the magnetic patterns regularly varied with the surface hardness of specimens, indicating the feasibility of surface hardness characterization with magnetic features.

In the second stage, the repeatability of the instrument under dynamic condition was concerned. During the repeatability test, the robotic arm gradually approached the specimen from a distance along the pre-planned trajectory. Once the sensor held at the end of the robotic arm contacted the specimen surface, it kept steady when performing multifunctional micro-magnetic tests ten times. Compared with the static condition, in the dynamic condition, the minor difference in contact state between the sensor and the specimen surface was included in the random uncertainties. The experiment was repeated 77 times and only performed on the specimen labeled as 4#. Therefore, a total of 770 signals were recorded to evaluate the repeatability of the instrument under dynamic conditions. Moreover, the data collected under dynamic conditions were used to evaluate the robustness of the trained prediction models in Section 4, considering the measured data of magnetic features (referred to as input nodes of the model) experienced slight fluctuations.

Multiple magnetic features were simultaneously measured with the instrument and thus the repeatability of the instrument should be discussed for individual magnetic features. Figure 3 demonstrates the measured data (770 data points sorted by the repeating times) of typical magnetic features. The error bars were plotted to demonstrate the extent of variation in the data under static conditions. The fluctuation in the mean value (red dot) as the repetitions indicated the repeatability of the instrument under dynamic conditions. The error bars were too short to recognize in Figure 3c,d, indicating the excellent repeatability of the instrument in measuring the magnetic features of *x*_20_ and *x*_36_ under static conditions.

As shown in Figure 3b, the measured data of magnetic features of *x*_15_ (extracted from the MBN signal) showed severe fluctuations during every test. Through recalling the measured patterns of MBN, the MBN envelope of a half magnetization cycle is demonstrated as the 3D image in Figure 4a. Compared with the smooth image of EIP in Figure 4b, both the peak and its position of MBN envelop vary in a certain range to cause a noised ridge along the axis of date. The distortion in the MBN envelop observed under dynamic conditions was mainly ascribed to the randomness of the Barkhausen events, which had been proven as a stochastic process during the material magnetization [18]. Therefore, the repeatability of the instrument in measuring magnetic features extracted from MBN signals was expected to be worse than that of the magnetic features of EIP.

## 3. Repeatability of the Instrument

The physical meanings and dimensions of all the measured 41 magnetic features were different from each other. A dimensionless index, coefficient of variation (*β*) estimated from the repeatability test data, was used to evaluate the repeatability of the instrument in measuring different magnetic features at the same scale. The coefficient of variation is defined as the ratio of standard deviation to the mean value. The repeatability of the instrument is discussed under static and dynamic conditions. The coefficient of variation of all the 41 magnetic features obtained from specimen 4# under static conditions was estimated and drawn as the curved surface in Figure 5a. The dispersion of *β* estimated from the 77 times of repeatability tests was plotted as the error bars in Figure 5b.

Several magnetic features of MFEC and the magnetic features of *x*_3_ and *x*_6_ (extracted from the 5th harmonic of TMF) demonstrates the high values of *β* (greater than 8%) due to the high-frequency magnetization of the tested specimen. During the high-frequency magnetization process, the magnetic attraction force between the yoke and the specimen might experience high-frequency variations and cause micro-vibrations of the sensor. The micro-vibration that occurred at the interface between the sensor and the specimen surface negatively affected MFEC which was very sensitive to the variation in the lift-off of the sensor.

For the magnetic features of EIP, its corresponding coefficient of variation (*β*) experienced very slight variations. For instance, the values of *β* corresponding to magnetic features of *x*_20_ and *x*_25_ vary in the ranges of 0.017~0.077% and 0.020~0.14%, respectively. Though the MBN features demonstrated larger values of *β* than those of EIP, the values of *β* remained basically stable and the coefficient of variation was less than 4%. Therefore, the repeatability of the instrument under static conditions is good, especially for measuring the magnetic features of EIP.

The repeatability of the instrument under dynamic conditions was estimated with the variations of the mean values of magnetic features measured under dynamic conditions. The coefficient of variation estimated from the data of the mean value of the magnetic feature, which can be observed from the data points marked as red dots in Figure 3a–d, is shown as the histogram in Figure 6. As expected, the magnetic features with a large value of *β* under static conditions also demonstrated significant fluctuations under dynamic conditions. However, under dynamic conditions, the values of *β* corresponding to the magnetic features of MBN and EIP are very close, which differs from the conclusions observed under static conditions. 

## 4. Establishment of the Prediction Models

### 4.1. Feature Selection

In the experiment, the observed variation in magnetic features was mainly ascribed to the change in surface hardness and the random error related to the repeatability of the instrument. For each magnetic feature of *x_i_* (*i* = 1, 2, …, 41), ten repeated test data were collected from the *j*_th_ (*j* = 1, 2, …, 8) specimen. The averaged value of *x_i_* is denoted as *m_ij_*. The variation range of *x_i_* is recorded as *C_i_*, which can be expressed as:(1)$$C_{i}=\max\{m_{ij}\}-\min\{m_{ij}\}$$

The magnetic feature *x_i_* of the *k*th (*k* = 1, 2, …, 10) test performed with the *j*th specimen is represented as *n_ijk_* and the random error of *x_i_* in the repeatability test is given as *P_ij_*:(2)$$P_{ij}=\max\{n_{ijk}\}-\min\{n_{ijk}\}$$

Therefore, the averaged random error of the feature *x_i_* can be expressed as:(3)$$\bar{P_{i}}=\left[\sum_{j=1}^{N}P_{ij}\right]/N$$
where *N* represents the quantity of specimens. To estimate the performance of magnetic features in surface hardness evaluation, an indicator similar to the signal-to-noise ratio is proposed as $F_{i}=\bar{P_{i}}/C_{i}$. The estimated values of *F_i_* are plotted as bars in Figure 7. When $F_{i}<5$, the corresponding magnetic feature of *x_i_* is considered to be not enough for surface hardness characterization.

Both the performance of magnetic features in surface hardness evaluation and the repeatability of the instrument are key factors to be considered in the feature selection for prediction model training. The curve of 1/*F_i_* (Figure 7) and the curve of *β* (Figure 6) estimated under dynamic conditions indicated that most of the magnetic features with larger values of *β* (instability) also had a large value of 1/*F_i_* (poor performance in surface hardness characterization). However, several magnetic features (such as x3 and x6) had a small value of 1/Fi (good ability) but a large value of β (instability) and were not good options as the input nodes of the prediction model.

Considering that most of the magnetic features have a value of 1/*F_i_* lower than 0.2, 1/*F_i_* < 0.2 was selected as a threshold for the selection of magnetic features. Therefore, only those measured magnetic features which met the conditions of 1/Fi < 0.2 and β < 5% were selected as the input nodes of prediction models. The selected features are marked by the red dot below their symbols in Figure 7. The data of the filtered magnetic features obtained in the first stage together with the measured surface hardness listed in Table 1 were used to train the quantitative prediction models.

The dependencies of the selected magnetic features on the surface hardness are shown in Figure 8. The value of *x*_7_ is approximately linearly dependent on surface hardness, whereas the other three features (*x*_12_, *x*_19,_ and *x*_37_) demonstrate the nonlinear dependency on surface hardness.

### 4.2. Modelling Strategy

Many researchers reported the complicated and nonlinear correlations between multiple magnetic features (input nodes) and surface hardness (output node) of steels. Neuronal network-based models are recommended for surface hardness predictions due to their higher accuracy than the multiple linear regression models. However, the robustness evaluation of the neuronal network-based models in micro-magnetic testing was rarely reported. Therefore, the robustness of the prediction models will be clarified using the data from repeating test. 

The performance of a trained model is determined by the selected input nodes, the structure of the neural network and the training algorithm, etc. This study focused on the selection of input nodes of the model. The measured values of magnetic features changed in a certain range during repeatability tests under dynamic conditions even though the performance of the developed instrument was good (Figure 4). If the robustness of the mode was poor, minor fluctuations in the values of selected magnetic features (input nodes) might cause unacceptable errors in surface hardness prediction. 

To examine the impact of fluctuation in the input nodes on the prediction accuracy of the model, the mean impact value (MIV) algorithm is a promising option. Through actively applying variations in the values of input nodes, the changes in the output nodes of the model were estimated as the impact value. An input node with a large value of MIV had a high weight of influence on the model or demonstrated a strong correlation with the output node. From the perspective of sensitivity (or influence weight) evaluation, the magnetic features with high MIV were more suitable input nodes of the model. However, the selection of input nodes with larger values of MIV might decrease the model’s robustness because the definition of the MIV indicated the adaptability of the model to the fluctuation in an individual magnetic feature. Thus, the input nodes should be selected based on the consideration of the balance between sensitivity and robustness.

Removing partial magnetic features of higher MIV from the input nodes might enhance the robustness of prediction models at the cost of model accuracy. Inspired by this idea, a research strategy (Figure 9a) was proposed to obtain the model with a balanced prediction accuracy and robustness based on the consideration of the repeatability of the instrument.

Feed-forward neural network (FNN) was employed to map the correlations between multiple magnetic features (input nodes) and surface hardness (output node) of the tested specimens. Two hidden layers were selected for the FNN structure during the training process of all the models and the number of nodes at each hidden layer was selected as ten. All the FNN models were established in the platform of MATLAB with the training function of Levenberg-Marquardt. The finalized connection weight values among the nodes of a trained FNN were affected by the initially assigned weight values. In this study, the initial weight values among the nodes of FNN were assigned randomly to train numerous FNN models. The feasibility of the proposed input node selection method was evaluated with the generated numerous FNN models of different parameters.

The procedures for model training and robustness evaluation are described below.

***Step* 1:** Estimation of MIV for all the candidate input nodes

A total of one hundred FNN models were trained before evaluating the MIV of magnetic features (or candidate input nodes). During the training process of FNN models, the data obtained from the 4# specimen were excluded and the magnetic features measured from the other seven specimens in the first-stage experiment were used as the input dataset. Through estimating the MIV of individual magnetic features with the one hundred trained FNN models, all the investigated magnetic features could be sorted in descending order of their averaged MIV. The sorting results of candidate input nodes according to their estimated values of MIV are shown in Figure 9b.

***Step* 2:** Models established with filtered input nodes

In the first step, all the candidate input nodes were used for model training to generate a reference model. In the second step, according to the descending order of their averaged MIV, some candidate input nodes were rejected one by one and not used for model training. The number of the filtered input nodes is *k* and the case with *k* = 0 corresponds to the reference model. For the cases of different *k*, the process of model training was repeated by randomly assigning the initial connection weight values among the nodes. Among the trained models, a total of 1000 fine models with an internal validation error of less than ±5% were stored for further statistical analysis. The performance of typical FNN models employing different numbers of input nodes can be observed in Figure 9c in which the results of internal validation with the highest accuracy are demonstrated.

***Step* 3:** Robustness evaluation of established models

The magnetic features measured from the 4# specimen (repeated 77 times under dynamic conditions) were used to evaluate the robustness of each established model. External validation of the model accuracy was performed for each test and the mean absolute error (MAE) of the model in surface hardness prediction during 77 times of testing was recorded. The data of MAE estimated from all the 1000 models were used to plot the histogram. The skewness and median estimated from the histogram were used as the statistical indicators for model robustness evaluation. Large skewness and small median indicated the high robustness of the models with specific input nodes. Through comparing the robustness of models with *k* input nodes are filtered, the best scheme for input nodes selection will be determined. 

## 5. Robustness Evaluation of Prediction Models

According to the procedure sketched in Figure 9a, the FNN models with *k*-filtered magnetic features were generated. The prediction accuracy of the established models with different values of *k* could be evaluated with the histograms (Figure 10). For the reference model (*k* = 0), its corresponding median and sknewness were estimated as 15.3 HV and 1.04, respectively. When the value of *k* increased from 0 to 8, the median showed an overall descending trend, whereas the sknewness demonstrated an upward trend (Figure 11a). For instance, the median reached its minimum value of 11.0 HV among the investigated cases when *k* was 6. The value of sknewness obtained in the case of *k* = 6 was around 30% larger than that of the reference model, indicating an improvement in model robustness. 

In the range of *k* ≤ 6, an increase of *k* causes a decrease in the average value of MAE and an increase in the number of models with MAE less than 10. The results in Figure 11b prove that the probability of high accuracy and robustness of the model can be improved through rejecting the magnetic features of higher MIV considering the random uncertainties faced by the instrument. When k is greater than 6, the four curves in Figure 11 tend to be stable, indicating that the robustness of models could only be improved in a certain range of k. In addition, the effective range of *k* depended on the investigated cases and should be further discussed based on the proposed modeling strategy.

To clearly display the improvement in the robustness of the prediction model, the model with the highest accuracy among the trained ones is recalled to predict the surface hardness of specimen 4# with the data of repeatability tests. For the cases with different values of *k*, the quantitative prediction results of the best models are shown in Figure 12. The surface hardness of the specimen was a constant and the dispersion of the prediction data near its true value was caused by the uncertainties during the measurement process of magnetic features. With the increase in *k* value, the extent of dispersion of the predicted data decreased.

In the traditional way (*k* = 0), all the magnetic parameters were used as the input nodes of model, and the MAE value of the model was about 2.42 HV (0.621%). The mean value of the predicted surface hardness had a negative offset of around 2.22 HV from its true value (Figure 12a), causing an underestimation of the surface hardness in most cases. When four magnetic features with a large MIV value were removed from input nodes, the prediction accuracy of the corresponding model (*k* = 4) was improved and the mean value was around 388.9 HV, which was closer to its true value of 389 HV. The MAE value and standard deviation of the results predicted by the model of *k* = 4 (Figure 12b) were estimated as 1.13 HV (around 0.291% of its true value) and 1.41 HV, respectively. The performance of the model could be further improved by increasing the value of *k* from 4 to 8. With the results in Figure 12c, the MAE value and standard deviation in surface hardness prediction were calculated to be around 1.02 HV (around 0.291% of its true value) and 1.04 HV, respectively. The reduction in both the MAE value and the data dispersion extent clearly verified that the robustness of the model was improved with the proposed modeling strategy. The multi-functional micromagnetic instrument equipped with the robust prediction model is expected to improve the quantitative prediction performances of surface hardness although unavoidable random uncertainties are involved in the test.

## 6. Conclusions

Multifunctional micromagnetic tests were performed with the specimens of Cr12MoV steel in order to realize the robust and quantitative prediction of surface hardness. The repeatability of the instrument in measuring magnetic features is imperfect and also may suffer random fluctuation during repeatability tests, thus resulting in changes in the input nodes of prediction models. The models of high robustness may be successfully applied in multifunctional micromagnetic testing. In this study, a modeling strategy and the selection method of input nodes were proposed for improving the robustness of FNN models in predicting the surface hardness of Cr12MoV steel. The conclusions are drawn as follows:The evaluation results obtained under both static and dynamic conditions verified that the self-developed multifunctional micromagnetic instrument had good repeatability in measuring the magnetic features of EIP and MBN and most of the magnetic features of TMF.Through the indicator combination of *F_i_* and *β*, magnetic features of good repeatability and good performance in surface hardness evaluation could be selected. However, the selection of input nodes from the filtered magnetic features should be further explored. The magnetic features with high MIV negatively affected the robustness of prediction models.Based on the consideration of the balance between the MIV of input nodes and the robustness of prediction models, removing partial magnetic features of high MIV from the input nodes could improve the robustness of prediction models.

## Figures and Tables

**Figure 1 sensors-23-01273-f001:**
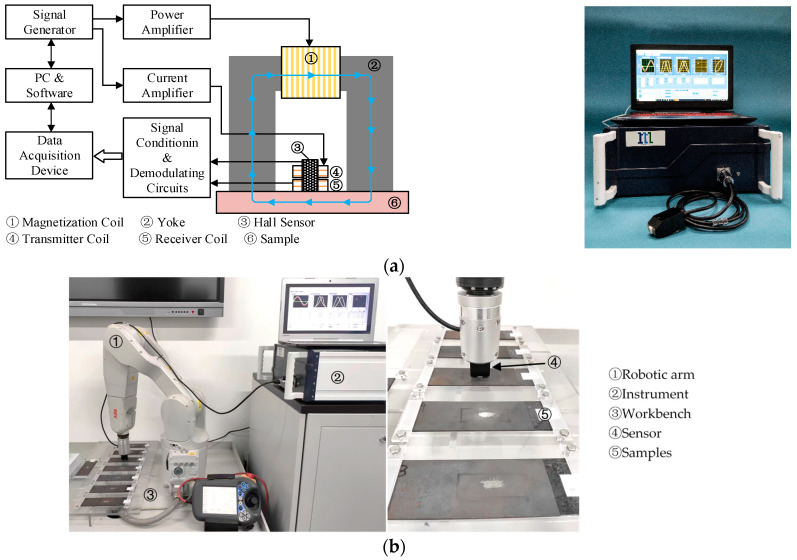
(**a**) Instrument and (**b**) experimental set-up for performing multifunctional micro-magnetic testing.

**Figure 2 sensors-23-01273-f002:**
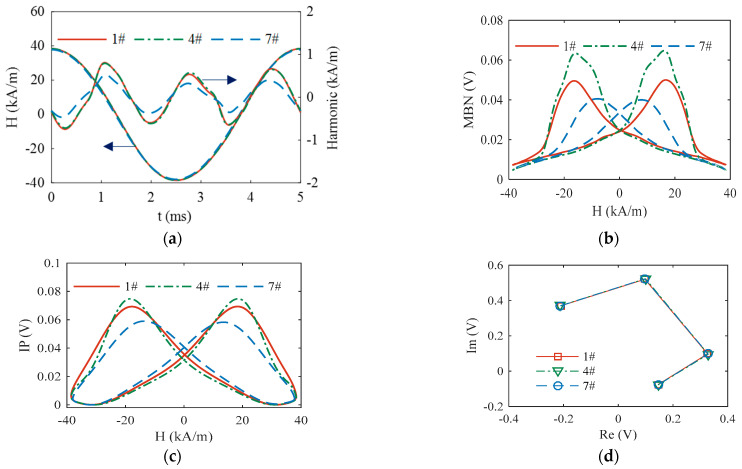
(**a**–**d**) demonstrate the patterns of TMF, MBN, EIP, and MFEC, respectively.

**Figure 3 sensors-23-01273-f003:**
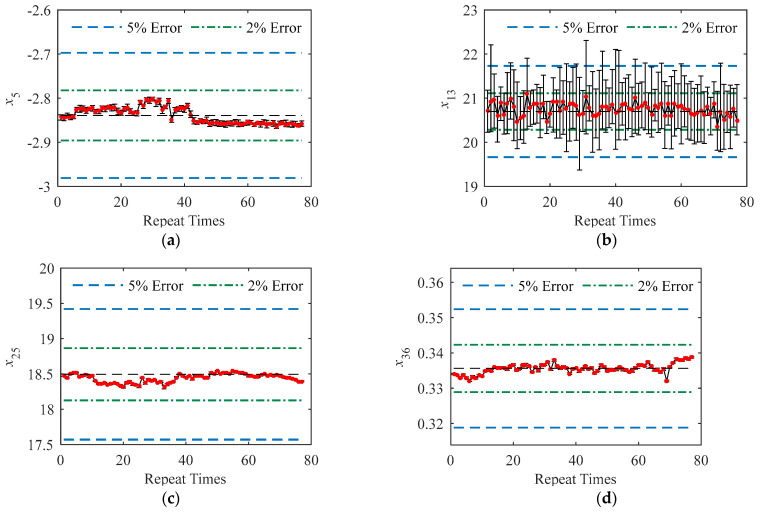
(**a**–**d**) demonstrate the recorded data of *x*_5_, *x*_15_, *x*_20,_ and *x*_36_, respectively.

**Figure 4 sensors-23-01273-f004:**
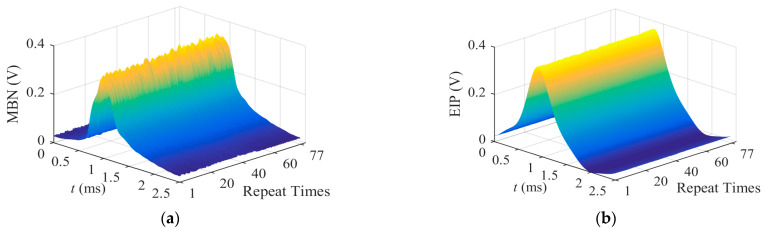
The 3D images of (**a**) MBN and (**b**) EIP envelop recorded in repeatability tests.

**Figure 5 sensors-23-01273-f005:**
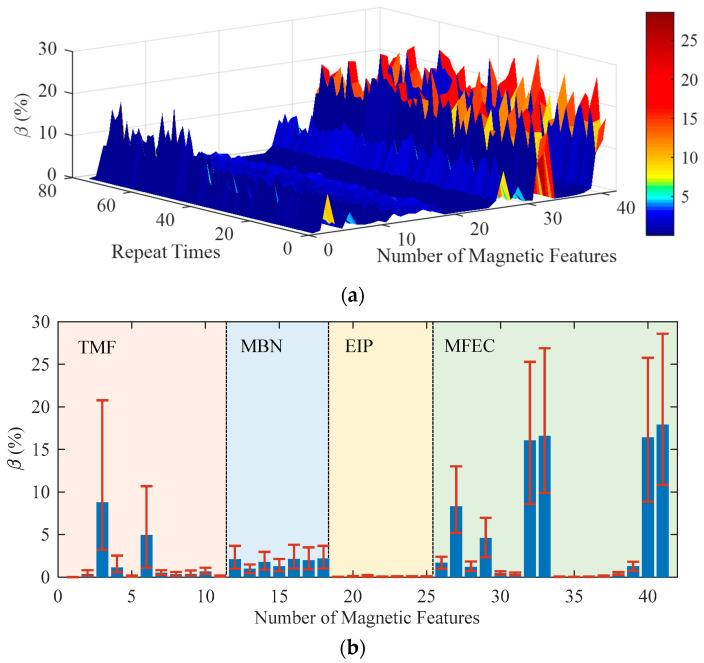
The coefficient of variation of the magnetic features estimated under static conditions. (**a**) and (**b**) demonstrate the estimated results in 3D and 1D, respectively.

**Figure 6 sensors-23-01273-f006:**
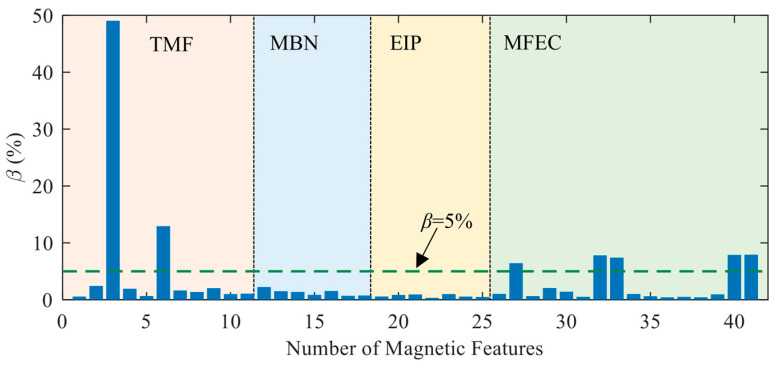
Estimated coefficients of variations of the magnetic features under dynamic conditions.

**Figure 7 sensors-23-01273-f007:**
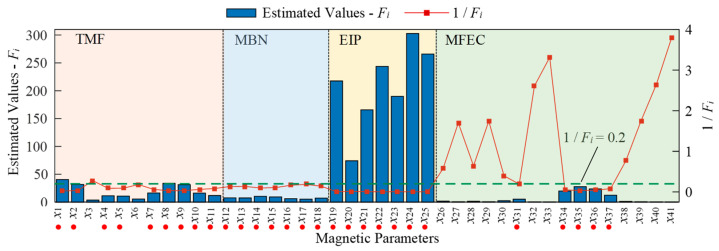
Estimated values of *F_i_* and 1/*F_i_* for all the measured magnetic features.

**Figure 8 sensors-23-01273-f008:**
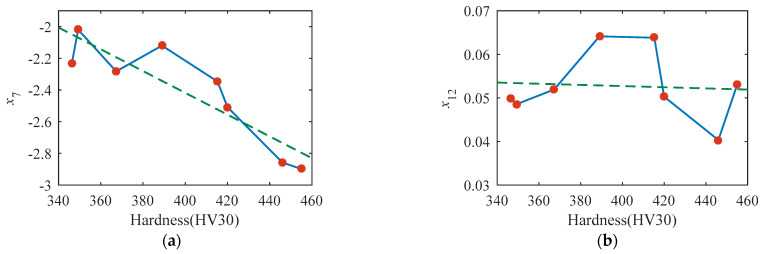
(**a**–**d**) demonstrate the dependency of *x*_7_, *x*_12_, *x*_19,_ and *x*_37_ on the surface hardness, respectively.

**Figure 9 sensors-23-01273-f009:**
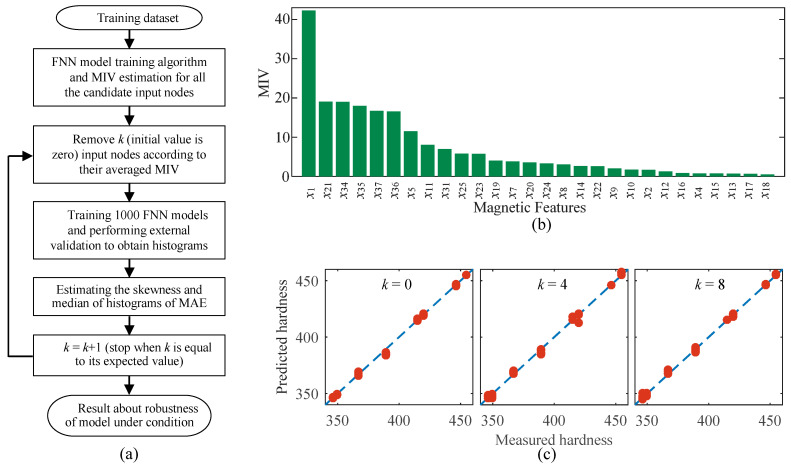
(**a**) Flow chart for the robustness evaluation of established models. (**b**,**c**) Magnetic features sorted by their estimated MIV values and performances of typical FNN models, respectively.

**Figure 10 sensors-23-01273-f010:**
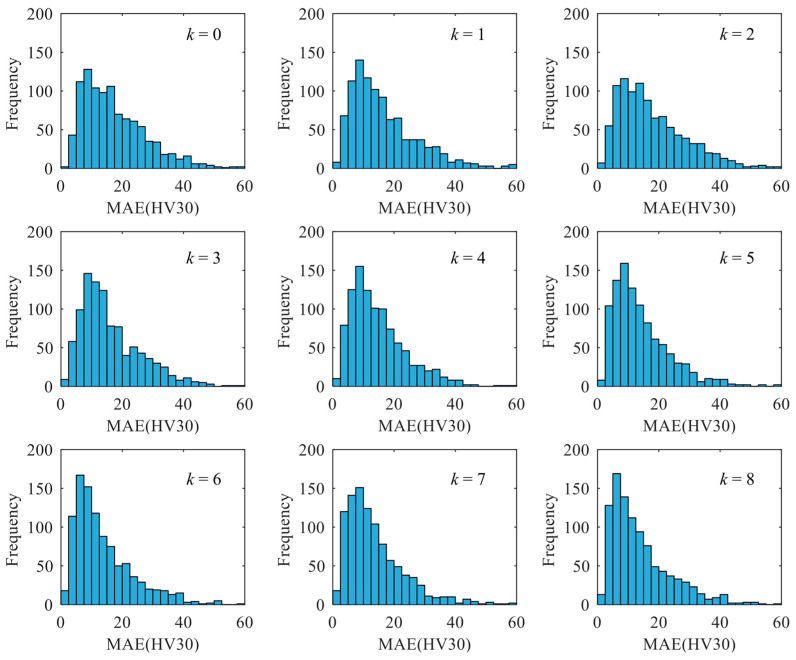
Histograms of MAE obtained from FNN models with different *k*.

**Figure 11 sensors-23-01273-f011:**
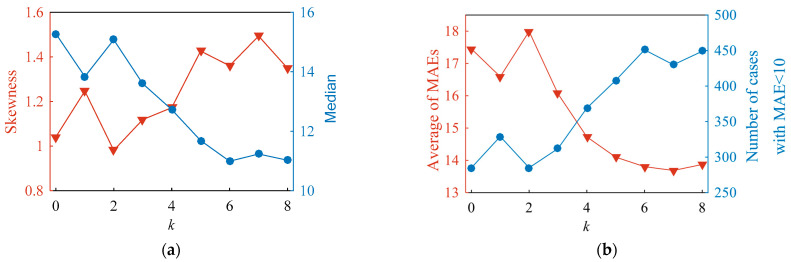
(**a**) Curves of estimated skewness and median; (**b**) Average MAE and the number of cases with MAE less than 10.

**Figure 12 sensors-23-01273-f012:**
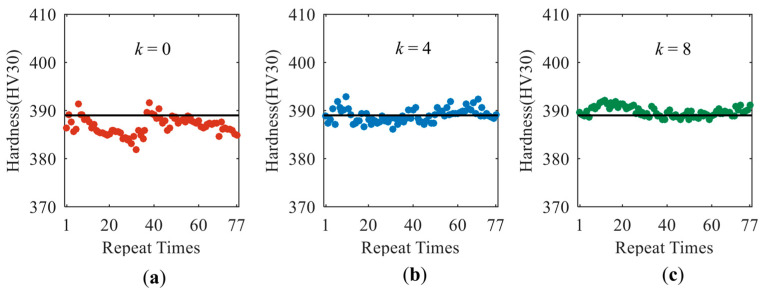
(**a**–**c**) show the quantitative prediction results of surface hardness with the models of *k* = 0, *k* = 4 and *k* = 8, respectively.

**Table 1 sensors-23-01273-t001:** The measured surface hardness of the prepared samples.

Specimen Nos.	Hardness (HV30)		
	Position 1	Position 2	Position 3
1#	347	346	346
2#	350	349	349
3#	369	367	366
4#	392	386	389
5#	414	416	416
6#	420	420	420
7#	448	444	446
8#	457	453	455

## Data Availability

The data presented in this study are available on request from the corresponding author.

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
