# Peer review of "Micromagnetic and Robust Evaluation of Surface Hardness in Cr12MoV Steel Considering Repeatability of the Instrument"

_sensors, 2023, doi:10.3390/s23031273_

Round 1

Reviewer 1 Report

In this work, it has been tried to correlate micromagnetic testing as a NDT with surface hardness of Cr12MoV steel focusing on the prediction accuracy considering the repeatability of the instrument using a neural network prediction model.  The authors have done some experimental works; however, some important scientific part of this manuscript is missing. Some comments are hereafter listed.

1. Manuscript need some grammatical check by an expert.

2. The manuscript is not well-written in some parts and need to be re-organized, particularly in the modelling strategy and its correlation with results and discussion section.

3. Novelty of work is significant and partially well supported by results as the repeatability but not for prediction models of high robustness, can authors explain how this measurements can be conducted on other alloys? Also, is there any significance on the roughness of the samples tested?

4. In this work, microstructural features: type of alloy, magnetic properties of an alloy and metallurgical conditions can in particular correlated with material magnetization, has these properties or the anisotropy of a sample an effect on measurements?

Author Response

Response to Reviewer 1 Comments

We feel great thanks for your professional review work on our article. According to your suggestions, we have made corrections to our previous draft, the details are listed below.

Q1. Manuscript need some grammatical check by an expert.

Response:

We ask help from the company of Tuopu Huawen Translation Co., Ltd, which is specialized in academic paper polishing, to improve the language of our paper. The records of revision can be seen in the revised version of paper.

Q2. The manuscript is not well-written in some parts and need to be re-organized, particularly in the modelling strategy and its correlation with results and discussion section.

Response:

Thank you for the suggestions. Indeed the structure of the original Section 4 is not well-organized and it is not easy to find the correlation between the modelling strategy and the results and discussion section.

In the revised paper, the title of Section 4 is revised as “Establishment of the prediction models” and the original section 4.3 “results and discussion” is changed as Section 5 “Robustness evaluation of prediction models”. Therefore, in Section 4 we focus on illustrating how to establish the prediction models and to evaluate their robustness. In Section 5 we show the results about the robustness evaluation of prediction models.

Q3. Novelty of work is significant and partially well supported by results as the repeatability but not for prediction models of high robustness, can authors explain how this measurements can be conducted on other alloys? Also, is there any significance on the roughness of the samples tested?

Response:

The micromagnetic technology is only applicable to ferromagnetic materials. The multifunctional micromagnetic testing proposed in the paper is applicable for other alloys that are ferromagnetic. This is common sense in the community of micromagnetic NDT technology. The measurements about repeatability of instrument just follows the general procedure as defined. The novelty of our work is about how to establish prediction models of high robustness considering repeatability of instrument. The proposed modelling strategy is applicable for other cases while high robustness models considering repeatability of instrument are desired.

The surface roughness has impact on the micro-magnetic signals. However, the test samples have been polished to have nearly the same condition of surface. In addition, the detection position at the tested 4# sample does not change during the experiment. Therefore, the effect of roughness of the sample is not be considered in this paper.

Q4. In this work, microstructural features: type of alloy, magnetic properties of an alloy and metallurgical conditions can in particular correlated with material magnetization, has these properties or the anisotropy of a sample an effect on measurements?

Response:

Yes, it is correct. The microstructures of ferromagnetic materials are key factors correlated with material magnetization. This is related with the principle of micromagnetic NDT technology. The principle of micromagnetic NDT technology has been described in the Introduction part as follows,

In ferromagnetic materials, the microstructures (grain boundary, precipitates, dislocation, etc.) not only hinder the motion of magnetic domains under the action of an external magnetic field but also determine the mechanical behavior under an external load [1]. Intrinsic connections between the magnetic properties and mechanical properties of ferromagnetic materials had been confirmed by numerous experiments [2-5]. Such phenomenological conclusions suggest the feasibility of magnetic evaluation of mechanical properties in nondestructive ways.

Reviewer 2 Report

The manuscript, entitled " Micromagnetic and Robust Evaluation of Surface Hardness in Cr12MoV Steel Considering Repeatability of the Instrument" is interesting and has some value. The presented results could be of interest for the reader of Sensors. It is well-written, readable and appropriately referenced. Therefore, my overall recommendation is “minor revision”. Further improvement is required as list below.

1.      In line 167, the author divides the experiment into static and dynamic conditions. However, dynamic conditions will include static conditions, so theoretically the repeatability of experimental results under dynamic conditions should be lower than that under static conditions. The authors present counter-intuitive results in Figures 5 and 6, which need further clarification here.

2.   The author should clearly describe the meaning of Fi to avoid the reader's misunderstanding.

3.  In Figure 7, when Fi equals 50, 1/ Fi should equal 0.02 versus 0.2. In addition, why was the threshold chosen?

4. The authors believe that the performance of the model can be improved by increasing the value of k, so what should be taken from k? In other words, what happens when k gets bigger?

Author Response

Response to Reviewer 2 Comments

We feel great thanks for your professional review work on our article. According to your suggestions, we have made corrections to our previous draft, the details are listed below.

Q1. In line 167, the author divides the experiment into static and dynamic conditions. However, dynamic conditions will include static conditions, so theoretically the repeatability of experimental results under dynamic conditions should be lower than that under static conditions. The authors present counter-intuitive results in Figures 5 and 6, which need further clarification here.

Response:

Under static condition, the sources of random error faced by the instrument mainly originate from the stochastic process of domain motion and slight fluctuation in electrical parameters of instrument. As compared with the static condition, in dynamic condition the minor difference of contact state between the sensor and the specimen’s surface is included in the random uncertainties. The contact state between the sensor and the specimen’s surface is related with the movement of robot. Therefore, the repeatability of experimental results under dynamic conditions is higher than that is obtained under static condition.

Q2.The author should clearly describe the meaning of Fi to avoid the reader's misunderstanding.

Response:

The definiation of Fi is in detail illustrated at the beginning of Section 4.1. As stated in the senstence below Eq.(3), an indicator similar to signal-to-noise ratio is proposed as .

Q3. In Figure 7, when Fi equals 50, 1/ Fi should equal 0.02 versus 0.2. In addition, why was the threshold chosen?

Response:

We carefully check the data and we confirm the value of 1/Fi is correct. The readers may feel confused about the value of 1/Fi because the value of 1/ Fi of different magnetic features varies in a very large range (0~4). Low value of 1/ Fi is preferred and 1/Fi = 0.2 is selected as threshold considering that most of the magnetic features has a value of 1/ Fi lower than 0.2. According we add an explanation to the related part as follows in the paragraph 4, section 4.1,

Considering that most of the magnetic features has a value of 1/ Fi lower than 0.2, 1/Fi<0.2 was selected as a threshold for the selection of magnetic features.

Q4. The authors believe that the performance of the model can be improved by increasing the value of k, so what should be taken from k? In other words, what happens when k gets bigger?

Response:

When the value of k increases from 0 to 8, the robustness of FNN model is improved as demonstrated in Figure 11. It is noticed that when k is greater than 6, the four curves in Figure 11 tend to be stable, indicating that the robustness of models could only be improved in a certain range of k. In addition, the effective range of k depended on the investigated cases and should be further discussed based on the proposed modelling strategy. The above statement is added to the related part in paragraph 2, section 5.
